# C-Type Natriuretic Peptide (CNP) Could Improve Sperm Motility and Reproductive Function of Asthenozoospermia

**DOI:** 10.3390/ijms231810370

**Published:** 2022-09-08

**Authors:** Na Li, Xinyi Dong, Sen Fu, Xiaoyan Wang, Huaibiao Li, Ge Song, Donghui Huang

**Affiliations:** 1Institute of Reproduction Health Research, Tongji Medical College, Huazhong University of Science and Technology, Wuhan 430030, China; 2Reproductive Center, Qingdao Women and Children’s Hospital Affiliated to Qingdao University, Qingdao 266034, China; 3NHC Key Laboratory of Male Reproduction and Genetics, Family Planning Research Institute of Guangdong Province, Guangzhou 510006, China; 4Shenzhen Huazhong University of Science and Technology Research Institute, Huazhong University of Science and Technology, Shenzhen 518109, China

**Keywords:** CNP, asthenozoospermia, sperm motility, male infertility, oxidative stress

## Abstract

This study is to analyze the effect of C-type natriuretic peptide (CNP) on sperm motility of asthenozoospermia and explore the influence mechanism of CNP on the reproductive system and sperm motility. Our results showed that the concentration of CNP in asthenospermia patients’ semen was lower than in normal people’s. The motility of sperm could be improved markedly by CNP and 8-Br-cGMP, while the effect of CNP was inhibited by NPR-B antagonist and KT5823. In the asthenozoospermia mouse model induced by CTX, CNP injection could improve sperm motility in the epididymis, alleviate tissue damage in the testes and epididymis, and increase testosterone levels. The asthenospermia mouse model showed high activity of MDA and proinflammatory factors (TNF-α, IL-6), as well as low expression of antioxidants (SOD, GSH-Px, CAT) in the testis and epididymis, but this situation could be significantly ameliorated after being treated with CNP. Those studies indicated that the concentration of CNP in the semen of asthenospermia patients is lower than in normal people and could significantly promote sperm motility through the NPR-B/cGMP pathway. In the asthenospermia mouse model induced by CTX, CNP can alleviate the damage of cyclophosphamide to the reproductive system and sperm motility. The mechanism may involve increasing testosterone and reducing ROS and proinflammatory factors to damage the tissue and sperm.

## 1. Introduction

According to relevant research statistics, there are about 15% of couples of childbearing ages who are infertile in the world, of which the male factor accounts for about 50% [1,2]. Asthenospermia, also known as low sperm motility, is defined as a condition in which the percentage of motility sperm in ejaculated semen is less than 40%, or the percentage of forward-moving sperm is less than 32% [3]. The sperm motility is directly related to the ability of male reproduction, which is necessary to ensure sperm reach the ampulla of the fallopian tube and entry eggs to form fertilized eggs. A retrospective study found that asthenozoospermia and teratozoospermia accounted for 50.5% and 54.1% in all infertile patients, respectively [4]. Therefore, asthenospermia is one of the main causes of male infertility.

Seminal plasma is a complex mixture of secretions from various glands in the male reproductive tract (such as seminal vesicle, prostate, urethral bulbar gland, epididymis, testes, etc.) [5], which mainly includes ions, lipids, proteins, enzymes, and sugars, and is responsible for sperm motility, fertilization events and so on [5,6]. Narciandi F. et al. demonstrated that high expression of β-defensins has been found in the seminal fluid and has been shown to affect sperm motility [7]. Zheng P. et al. reported that uPA in semen improved sperm mobility, induced AR, and enhanced sperm capacity to fertilize mature eggs [8]. Application of testosterone supplementation in semen to improve sperm motility in asthenozoospermia males [9]. Therefore, the abnormality or absence of seminal plasma components will lead to the insufficiency of sperm quality.

In the 1990s, an unknown peptide with an amino acid sequence significantly similar to an atrial natriuretic peptide (ANP) and brain natriuretic peptide (BNP) was found in pig brain extract and named C-type natriuretic peptide (CNP) [10]. Chrisman et al. detected that the concentration of CNP in pig seminal plasma was 2000 times that in pig brain tissue [11]. Nielsen et al. found that the protein level of CNP in human seminal plasma is 200 times that in plasma [12]. Furthermore, Nielsen et al. also showed that both CNP and its precursor, Pro-CNP, were highly expressed in the male reproductive system (mainly epididymis, seminal vesicle, and prostate) [13]. All of these indicated the important role of CNP in sperm function. Kong et al. found that CNP was highly expressed in the oviduct epithelium of the ampullary of mice and attracted sperm toward oocytes by increasing intracellular levels of cGMP and Ca^2+^ of spermatozoa, which was essential for fertilization [14]. Our previous studies showed that CNP could enhance sperm motility [15], stimulate intracellular cGMP/PKG signal transduction, increase Ca^2+^ influx and tyrosine protein phosphorylation, induce sperm hyperactivation and acrosome reaction, and finally promote sperm capacitation [16].

The above results indicate that CNP is closely related to sperm motility. The expression of CNP in the testis of the rat infertile model induced by ornidazole decreased than that in normal rat [17]. Tomasiuk et al. indicated that CNP may be a new marker of asthenospermia [18]. In order to further study the relationship between CNP and asthenospermia, this research detected the effect of CNP on sperm function in asthenospermia patients and the established asthenospermia animal models to explore possible mechanisms.

## 2. Results

### 2.1. CNP Levels in Semen of Normal People and Patients with Asthenospermia

A total of 47 cases of normal semen samples and 45 cases of asthenospermia semen samples were collected, and the concentrations of CNP in the semen of normal people were lower than those in the semen of patients with asthenospermia (*p* < 0.01) (Table 1).

### 2.2. The Effect of CNP on Sperm Motility of Asthenospermia Patients In Vitro

In asthenozoospermia patients, the motility of sperm treated with CNP (10^−6^ mol/L) is higher than that untreated with CNP in vitro, especially after 60 min and 120 min (*p* < 0.01) (Figure 1a). After 1 h of treatment, the sperm motilities in the CNP group and 8-Br-cGMP group were higher than those in the control group (*p* < 0.01), while the sperm motilities in CNP + KT5823 group and NPR-B antagonist group were lower than that in the CNP group (*p* < 0.01) (Figure 1b). Meanwhile, the concentrations of cGMP in sperm of each group were detected, and the results showed that treads of cGMP were consistent with the trend of sperm motility (Figure 1c). 

### 2.3. The Effects of CNP on Reproductive System in Asthenospermia Mouse Model

In the asthenospermia mouse model induced by CTX, the body weight, epididymis index, sperm motility (PR + NP), and sperm concentration were significantly lower than those in control groups (*p* < 0.05); after being treated with CNP, the body weight, epididymis index, sperm motility (PR + NP) and sperm concentration were significantly improved than those in CTX groups (*p* < 0.05) (Table 2). Hematoxylin-eosin staining showed that the spermatogonial tubule lumen of the testis in the CTX group was enlarged, the spermatogonial cells were deformed and vacuolated, the structural hierarchy was disordered, and the spermatozoa cells were arranged disorderly (Figure 2(Bd)); meanwhile, the lumen of epididymis was enlarged, sperm cells were reduced, the structure of epithelial cell was damaged, and interstitial space was enlarged (Figure 2(Be)). In contrast, the tissue injury in the testis and epididymis in the CTX + CNP (25) group and CTX + CNP (50) group were alleviated to a certain extent than those in the CTX group. Testosterone levels in the CTX group decreased and were recovered partly in the CTX + CNP (25) group and CTX + CNP (50) group (Figure 2E). Furthermore, the effects of CNP (25) on reproductive function looked more obvious than that of CNP (50). 

### 2.4. The Effects of CNP on ROS Level in Asthenospermia Mouse Model

Immunofluorescence staining showed that ROS expression in the testis and epididymis in CTX group increased than that in control group (*p* < 0.01), while decreased than those in CTX + CNP (25) group and CTX + CNP (50) group (*p* < 0.01) (Figure 3). At the same time, CNP of different concentration has significant role in reducing ROS level. 

### 2.5. Effect of CNP on Immune Function in Asthenospermia Mouse Model

In CTX induced asthenospermia mouse model, the levels of TNF-α and IL-6 in serum were higher than those in the control group (*p* < 0.01). Moreover, CNP could alleviate the secretion of TNF-α and IL-6 (*p* < 0.01). The gene expression of TNF-α and IL-6 in the testis and epididymis showed the same treads (Figure 4). 

## 3. Discussion

This study was the first to find that CNP can improve sperm motility in asthenospermia patients by the cGMP signal pathway, and CNP can improve male reproductive function in asthenospermia model mice, which mechanism may be involved in regulating serum testosterone level, decreasing tissue oxidative stress level and reducing inflammatory reaction and so on.

There are many causes inducing asthenospermia, mainly including infection, abnormal chromosome or gene expression, endocrine factors, antisperm antibodies (immune factors), environmental and occupational exposure, sperm maturation disorders, varicocele, trace element or vitamin deficiency, abnormal DNA methylation, abnormal semen fluidization, oxidative stress level, iatrogenic and other factors [19,20,21]. Because the etiology of asthenospermia is complex and remains unclear, the treatment strategies for asthenospermia are relatively limited, such as increasing sperm motility, endocrine therapy, supplement of trace elements and vitamins, antioxidant drugs, etc. [21,22]. This experiment firstly examined the concentration of CNP in semen samples from patients with asthenospermia and found that the concentration of CNP in asthenospermia patients was lower than that of normal people. Moreover, CNP could increase sperm motility of asthenospermia patients in vitro, which suggests that CNP may be a potential treatment for asthenospermia patients. 

CNP is now regarded as an autocrine/paracrine regulator with broad expression in the body [23]. CNP produces a series of biological effects by the cGMP signaling pathway through binding to its specific natriuretic peptide receptor B (NPR-B) [24,25], such as regulation of the proliferation of myocardial cells [26]; maintaining cardiovascular homeostasis [27]; inhibition of bone growth and cartilage development [28]; promote follicular development [29]. Our group’s previous studies have already shown that CNP can induce sperm capacitation in normal people through cGMP [16]. In this experiment, we treated the sperm of asthenospermia patients with CNP, 8-Br-cGMP, KT5823, and NPR-B antagonist. The results showed that CNP and 8-Br-cGMP could increase sperm motility in asthenospermia patients, but the effect of CNP was inhibited by KT5823 and NPR-B antagonists. Moreover, cGMP concentration was consistent with the trend of sperm motility, which suggested that CNP increased asthenospermia’s sperm motility through the NPR-B/cGMP pathway.

In order to further study the mechanism of CNP’s effect on asthenospermia, we established an asthenospermia mouse model using CTX. CTX is an alkylation antitumor agent that is toxic to the male reproductive system [30]. Wang et al. used CTX to build an animal model and found that sperm motility decreased, presenting as asthenospermia [31]. Elangovan N et al. found that CTX could significantly reduce body weight and sperm motility, damage the integrity of the acrosomal structure, and decrease testosterone levels in male mice [32]. Our study showed that the body weight, gonad weight, and sperm motility in the CTX group were significantly lower than those in the control group, and these damages will be significantly relieved after treating with CNP, indicating that CNP can improve the growth status of asthenospermia mice and the damage to the reproductive system. We also found that CNP can promote the testosterone level in asthenospermia mice. El-Gehani F et al. reported that natriuretic peptides (ANP\BNP\CNP) all stimulated testosterone production, with significant effect at concentrations > or =1 × 10^−8^ mol/L of ANP, > or =1 × 10^−9^ mol/L of BNP, and > or =1 × 10^−6^ mol/L of CNP [33]. Knockdown of NPR2 by RNAi resulted in S phase cell cycle arrest, cell apoptosis, and decreased testosterone secretion in mouse Leydig cells [34]. Our results implied that CNP improved the male reproductive system to some extent through the regulation of testosterone levels.

Under normal conditions, sperm produce ROS to maintain their acrosomal response and sperm capacitation [35], but excessive ROS will cause lipid peroxidation of sperm, damage sperm DNA, affect sperm activity, and thus cause sperm dysfunction and affect male reproductive health [35,36]. Previous studies have shown that the activities of SOD and GSH-Px in the seminum of male infertility patients are significantly lower than those of normal men, while the levels of MDA and ROS are significantly higher than those of normal men [37]. Therefore, excessive ROS plays an important role in the treatment of sperm dysfunction, and lowering ROS can improve the fertility of patients with sperm dysfunction [38]. Zhenwei et al. demonstrated that CNP could increase the levels of GSH-Px and reduce the levels of ROS in bovine oocytes [39]. In this study, the testis and epididymis in the mouse asthenospermia model also showed high activity of ROS, and CNP could reverse the phenomenon, which indicated that CNP could reduce tissue oxidative stress levels and thus improve sperm motility.

Sperm hypofunction and poor motility are related to inflammatory processes caused by infections of the genital tract or gonads [40]. Proinflammatory factors (such as TNF-α and IL-6) in seminal plasma are closely related to sperm motility [41,42]. Kimura et al. reported that CNP significantly attenuated levels of proinflammatory factors (TNF-α, IL-1, IL-6, etc.) in acute lung injury mice induced by LPS [43]. Chen et al. found that CNP effectively attenuated LPS-induced endothelial activation by inhibiting the NF-κB and p38 signaling pathways [44]. Our previous studies also showed that CNP plays a protective anti-inflammatory effect in the epididymitis mouse model by NF-kB signal pathway, inhibiting NF-κB and P38 signaling pathways [45]. In this study, the mRNA expression levels of TNF-α and IL-6 were increased in the CTX group and decreased after being treated with CNP in a dose-dependent trend. These results indicated that CNP may improve male reproductive function by down-regulating proinflammatory factors.

In a word, CNP could not only promote sperm motility but also improve the male reproductive function in asthenospermia. The mechanism might be involved as follows: 1. promote the secretion of serum testosterone level, ameliorating endocrine of male reproduction; 2. increase the antioxidants and reduce ROS level in male reproductive organs; 3. decrease the expression of TNF-α and IL-6, protecting tissue from damage. The study is expected to provide a new clinical approach for asthenospermia patients.

## 4. Materials and Methods

### 4.1. Animals

Male C57BL/6 mice aged 6–8 weeks were purchased from the Hubei Center for Disease Control and Prevention and reared in the animal room of the Institute of Reproductive Health. The temperature in the animal room was controlled at about 20 °C, the relative humidity was about 70%, regular light and darkness were given for 12 h each, water and food were freely available, and the formal experiment was started after 3–5 days of adaptive feeding for all mice.

### 4.2. Human Sperm Samples

Normal human and asthenospermia semen samples were collected from the Reproduction Medicine Center of Tongji Medical College, Huazhong University of Science and Technology, and analyzed according to the 2010 World Health Organization semen reference [3]. The sperm samples were ejaculated into disposable sterile containers and liquefied for at least 30 min at 37 °C. Next, samples were processed by centrifugation for 5 min at 800× *g* to remove the seminal plasma and adjusted to 3 × 10^7^ spermatozoa/mL in Ham’s F-10 medium for each experiment. The seminal plasma was used to detect the CNP level with an ELISA kit (Elabscience Biotechnology Co., Ltd., Wuhan, China). This study was approved by the Ethical Committee of Tongji Medical College, Huazhong University of Science and Technology (No. S1188).

### 4.3. Determination of cGMP Levels in Human Spermatozoa by ELISA

Intracellular cGMP was extracted using the repeated freezing and thawing method. Briefly, after incubation, sperm samples were washed three times in PBS by centrifugation (500× *g*, 5 min), resuspended in 1 mL PBS in the frozen pipes, frozen in liquid nitrogen for 15 min, and thawed in a water bath at 37 °C for 5 min. The freeze–thaw process was repeated three times. Finally, the supernatant was used to determine the cGMP level with a cGMP Activity Assay Kit according to the manufacturer’s protocol (Biovol Technologies, Shanghai, China).

### 4.4. Establishment of Asthenospermia (Cyclophosphamide) Mouse Model

An animal model of asthenospermia was established by intraperitoneal injection of cyclophosphamide (CTX). The groups were divided into four groups as follows: ① Group A: blank control (PBS); ② Group B: CTX (50 mg/kg/d); ③ Group C: CTX (50 mg/kg/d) + CNP (25 μg/kg/d); ④ Group D: CTX (50 mg/kg/d) + CNP (50 μg/kg/d). The drug injection scheme was as follows: group A mice were injected with PBS for 10 days, once a day; Mice in group B were continuously injected with CTX (50 mg/kg/d) for 5 days, and then PBS for 5 days. Mice in group C were continuously injected CTX (50 mg/kg/d) for 5 days, and then CNP (25 μg/kg/d) for 5 days. Mice in group D were continuously injected with CTX (50 mg/kg/d) for 5 days and then CNP (50 μg/kg/d) for 5 days.

### 4.5. Statistics of Sperm Motility and Sperm Density

Sacrifice the mice by neck dislocation, aseptically separate the epididymis, and remove the surrounding mesangial membrane and fat. Place the sperm in an epididymis containing culture solution of Hams-F10, and collect the sperm for incubation (37 °C, 5% CO_2_, saturated humidity). Detection of sperm motility: take 10 μL sperm suspension specimen smear, and observe under the microscope for at least 5 fields continuously and repeat 3 times, count 200 sperm each repeat, record the percentage of forward motion (PR) and non-forward motion (NP) sperm, and count the sperm density by blood cell counting plate. Sperm density measurement: take 10 μL incubated sperm suspension specimen and place it on the blood counting plate. Count the number of sperm in four squares (each square is divided into 16 smaller squares) (only count intact and normal sperm).

### 4.6. Measurement of Serum Testosterone and Oxidative Stress Levels by ELISA

The mice were sacrificed by cervical dislocation, and the heart blood (about 0.4 mL) was collected from the mice. To collect supernatant after centrifugation and serum testosterone, TNF-αand IL-6 levels were detected by Mouse Testosterone ELISA Kit (Elabscience Biotechnology Co., Ltd., Wuhan, China).

### 4.7. Immunofluorescence

Detect tissue ROS level: (1) ROS Staining: draw a circle around the tissue with a histochemical pen, then incubate with ROS dye solution (Dihydroethidium, DHE; Wuhan Servicebio Technology Co., Ltd., Wuhan, China) for 30 min at 37 °C. (2) Counter-stained with DAPI: slides were washed three times (5 min each wash), then counter-stained with DAPI for 10 min at room temperature. (3) Mounting: slides were washed as before and mounted with antifade mounting medium. (4) Microscopic photography: observe and collect images under a fluorescence microscope.

### 4.8. Statistical Analyses

All data were statistically analyzed using SPSS software, Version 23.0 (IBM Corp., Armonk, NY, USA). The means ± standard deviations (SD) and 95% confidence intervals (CI) were reported for quantitative data, and percentages were reported for categorical data. A level of *p* < 0.05 was considered statistically significant.

## Figures and Tables

**Figure 1 ijms-23-10370-f001:**
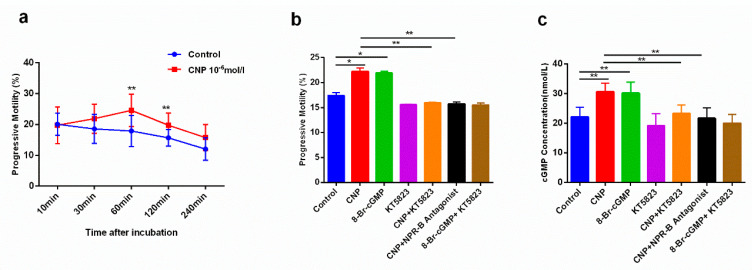
CNP could improve sperm motility in asthenospermia patients by cGMP signal. (**a**): CNP could improve sperm motility at different times, especially at 60 min, 120 min; (**b**): The effect of CNP on sperm motility was inhibited by KT5823 and NPR-B antagonist; (**c**): Treads of cGMP level in sperm were consistent with the trends of sperm motility. (Mean ± SD, *n* = 15, * *p* < 0.05,** *p* < 0.01).

**Figure 2 ijms-23-10370-f002:**
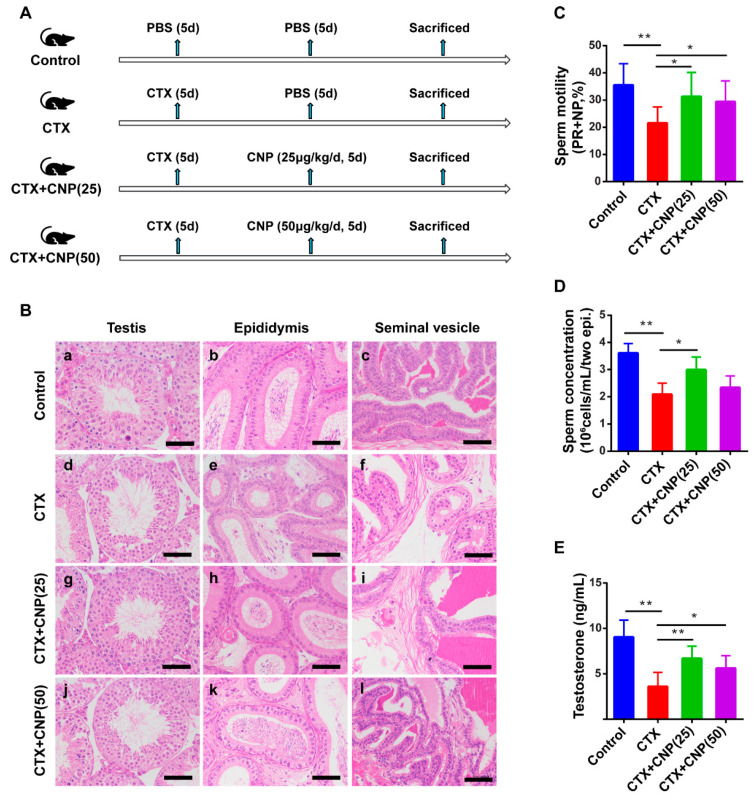
Effects of CNP on male reproductive function in the asthenospermia mouse model. (**A**). Experimental groups and steps to establish the model of asthenospermia. (**B**). Histological morphology of reproductive tissues. (**a**–**c**): Histological morphology of testis, epididymis, and seminal vesicle of mice in the control group were shown successively; (**d**–**f**): Histological morphology of testis, epididymis, and seminal vesicle in the CTX group; (**g**–**i**): Histological morphology of testis, epididymis, and seminal vesicle in the CTX + CNP (25) group; (**j**–**l**): Histological morphology of testis, epididymis and seminal vesicle in the CTX + CNP (50) group. Scale bars = 50 µm. (**C**): Effect of CNP on sperm motility in each group; (**D**): Effect of CNP on sperm density in each group; (**E**): Effect of CNP on serum testosterone levels. (* *p* < 0.05, ** *p* < 0.01).

**Figure 3 ijms-23-10370-f003:**
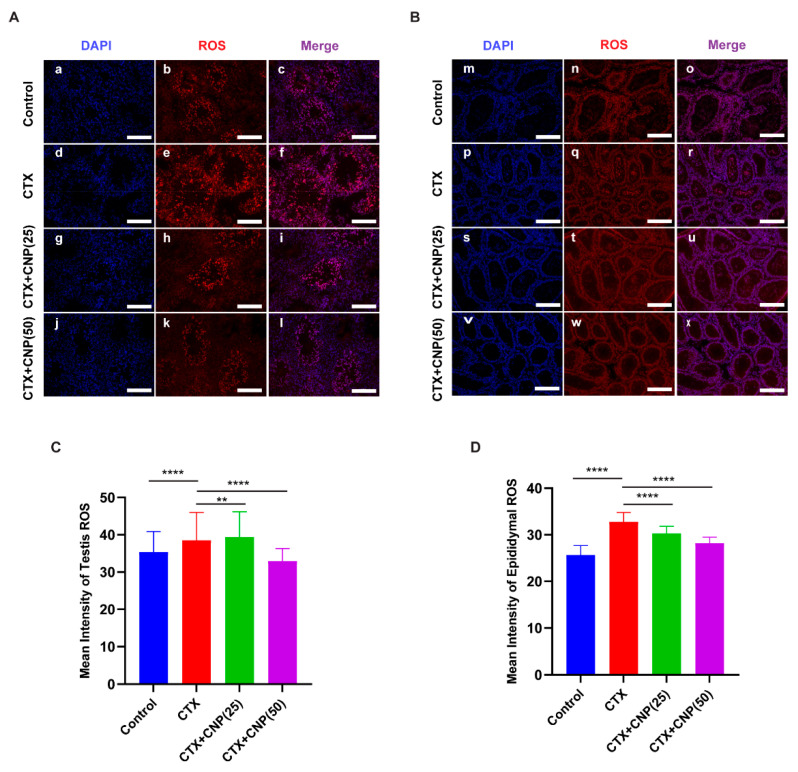
The effect of CNP on ROS expression in testis and epididymis of the asthenospermia mouse model. (**A**): Immunofluorescence of ROS in the testis of different groups. (**a**–**c**): Fluorescent microscopic images showing ROS of mouse testis in the control group; (**d**–**f**): Fluorescent microscopic images showing ROS of mouse testis in the CTX group; (**g**–**i**): Fluorescent microscopic images showing ROS of mouse testis in the CTX + CNP (25) group; (**j**–**l**): Fluorescent microscopic images showing ROS of mouse testis in the CTX + CNP (50) group. Scale bars = 200 µm. (**B**): Immunofluorescence of ROS in the epididymis of different groups; (**m**–**o**): Fluorescent microscopic images showing ROS of mouse epididymis in the control group; (**p**–**r**): Fluorescent microscopic images showing ROS of mouse epididymis in the CTX group; (**s**–**u**): Fluorescent microscopic images showing ROS of mouse epididymis in the CTX + CNP (25) group; (**v**–**x**): Fluorescent microscopic images showing ROS of mouse epididymis in the CTX + CNP (50) group. Scale bars = 200 µm. (**C**): Histogram showed the degree of ROS expression according to (**A**); (**D**): Histogram showed the degree of ROS expression according to (**B**). (** *p* < 0.01, **** *p* < 0.0001).

**Figure 4 ijms-23-10370-f004:**
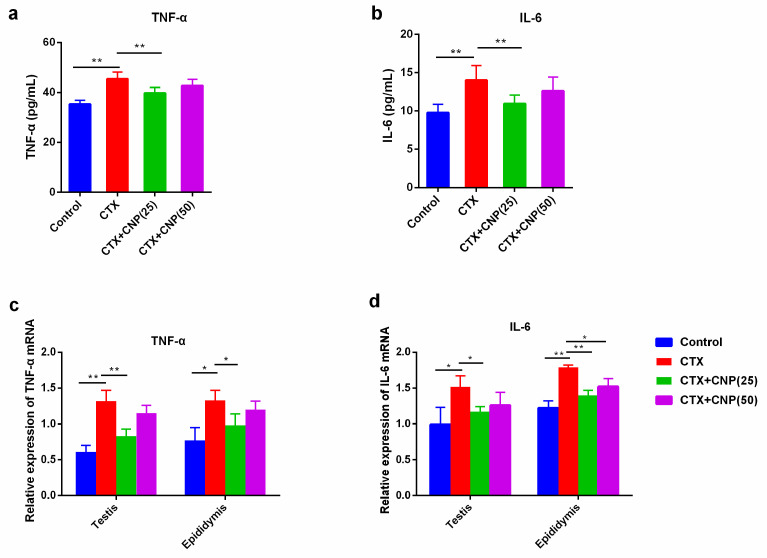
The effect of CNP on the expression of TNF-α and IL-6 in the asthenospermia mouse model. (**a**): The effect of CNP on serum TNF-α level in blood in each group; (**b**): The effect of CNP on serum IL-6 level in blood in each group; (**c**): The effect of CNP on TNF-α mRNA levels in testis and epididymis of each group; (**d**): The effect of CNP on IL-6 mRNA levels in testis and epididymis of each group. (* *p* < 0.05, ** *p* < 0.01).

**Table 1 ijms-23-10370-t001:** Basic clinical information of normal people and asthenospermia patients.

Basic Clinical Information	The Normal Group (*n* = 47)	The Asthenospermia Group (*n* = 45)
Age (years)	30.20 ± 3.97	29.16 ± 5.01
Semen volume (mL)	3.63 ± 0.74	3.51 ± 0.97
Sperm concentration (×10^6^/mL)	38.34 ± 11.64	27.79 ± 10.20
Total number of sperm (×10^6^)	135.80 ± 41.78	95.75 ± 36.79
Total motility (%)	67.32 ± 14.55	29.42 ± 6.43 **
VCL (μm/s)	71.98 ± 20.58	56.47 ± 19.79
VSL (μm/s)	44.15 ± 15.18	32.19 ± 10.74
VAP (μm/s)	47.74 ± 14.64	37.20 ± 12.80
CNP concentration (mean ± SD, pg/mL)	259.75 ± 12.57	223.04 ± 20.91 **

Note: ** compared with normal group, *p* < 0.01.

**Table 2 ijms-23-10370-t002:** Effects of CNP on body weight, gonad index, and semen quality of mice in each group.

	Control (*n* = 5)	CTX (*n* = 5)	CTX + CNP (25) (*n* = 5)	CTX + CNP (50) (*n* = 5)
Body weight (g)	24.22 ± 1.67	14.48 ± 1.78	17.04 ± 0.99 ^#^	15.45 ± 1.48
Testis weight (mg)	179.48 ± 9.48	104.50 ± 6.56 *	131.55 ± 17.79	117.57 ± 27.86
Testis index (mg/g b.w.)	7.41 ± 0.71	7.23 ± 0.49	7.72 ± 0.98	7.61 ± 1.07
Epididymis weight (mg)	53.30 ± 6.46	21.04 ± 9.30 *	34.59 ± 6.08	28.89 ± 13.72
Epididymis index (mg/g b.w.)	2.20 ± 0.25	1.45 ± 0.38 *	2.03 ± 0.34 ^#^	1.87 ± 0.29 ^#^
PR + NP (%)	30.50 ± 2.30	18.55 ± 3.42 *	24.25 ± 3.62 ^#^	22.81 ± 4.04 ^#^
Sperm concentration (10^6^ cells/mL/two epi.)	3.61 ± 0.35	2.09 ± 0.41 *	2.99 ± 0.48 ^#^	2.34 ± 0.43

Note: * compared with the control group, *p* < 0.05; ^#^ compared with the CTX group, *p* < 0.05.

## Data Availability

The data presented in this study are available on request from the corresponding author.

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
