# Peer review of "C-Type Natriuretic Peptide (CNP) Could Improve Sperm Motility and Reproductive Function of Asthenozoospermia"

_ijms, 2022, doi:10.3390/ijms231810370_

Round 1

Reviewer 1 Report

The paper does not bring a lot of Novelty. The redaction of the paper is rather careless.

A good example is given by the definition of CTX: this abbreviation is defined only in Mat and met: Cyclophosphamide: Line 248 !! However, it can be Chlorotoxin or Charybdotoxin

Lines 44-47: No, seminal plasma is an inhibitor of acrosome reaction. Seminal plasma contains also testicular fluid

Line 49 to 51: uPA? And then? Testosterone increases sperm quality…. LH , testosterone adult Leydig cells maintain male reproductive function by producing testosterone

Line 62: towards

In the text, mat and met: testosterone? It is obvious that if Testo increases in testis ,this is related to Leydig increase in

 Line 53 to 66, redaction neglected

Table 1: the lines 5,6,7, 8 must be removed, moreover undefined

Line 100: HE staining??

The two series of pictures:

Figure 2: the histology is not really obvious.

For testosterone

between line 122 and 123: unreadable

Mat and Met: line 274 to 284: totally un-understandable; not precise: what dye for ROS??

This paper is poorly written, a lot of info are missing in mat and met.

The block of phosphodiesterase and nothing really new

Author Response

Dear reviewer:

Thank you for your comments concerning our manuscript entitled “C- type natriuretic peptide (CNP) could improve sperm motility and reproductive function of asthenozoospermia” (ID: ijms-1860087). We have read these comments very carefully and found these suggestions are most helpful. Additionally, the manuscript was revised carefully according to the comments and marked in blue color. To be specific, we made revision as followed:

  1. A good example is given by the definition of CTX: this abbreviation is defined only in Mat and met: Cyclophosphamide: Line 248!! However, it can be Chlorotoxin or Charybdotoxin

Reply: Thanks for your suggestion. CTX is a common abbreviation of chemotherapeutic agents in clinical application, namely Cyclophosphamide. The aim of using CTX as abbreviation is to make picture easier.

  1. Lines 44-47: No, seminal plasma is an inhibitor of acrosome reaction. Seminal plasma contains also testicular fluid

Reply: Thanks for your good advices. We have added “testes” and deleted “acrosome reaction.  The sentences have changed to “Seminal plasma is a complex mixture of secretions from various glands in the male reproductive tract (such as seminal vesicle, prostate, urethral bulbar gland, epididymis, testes, etc.) [5], which mainly includes ions, lipids, proteins, enzymes, and sugars, and is responsible for sperm motility, fertilization events and so on”.

  1. Line 49 to 51: uPA? And then? Testosterone increases sperm quality…. LH , testosterone adult Leydig cells maintain male reproductive function by producing testosterone

Reply: Thanks for your suggestion. We added sentences to expound the importance of seminal plasma components, which is “Therefore, the abnormality or absence of seminal plasma components will lead to the insufficiency of sperm quality.”

  1. Line 62: towards

In the text, mat and met: testosterone? It is obvious that if Testo increases in testis ,this is related to Leydig increase in

Reply: I am very sorry, because I wouldn’t find the sentence and understand the meaning. Therefore, I couldn’t answer this question.

  1. Line 53 to 66, redaction neglected

Reply: Thanks for your suggestion. We have revised the mistakes.

  1. Table 1: the lines 5,6,7, 8 must be removed, moreover undefined

Reply: Thanks very much for your advice. We have removed the lines 6,7,8 and deleted PR+NP in the lines 5. Is it correct?

  1. Line 100: HE staining??

Reply: Thanks for your question. HE staining means hematoxylin-eosin staining, which has been revised in article and marked in blue.

  1. The two series of pictures:

Figure 2: the histology is not really obvious.

For testosterone

between line 122 and 123: unreadable

Reply: Thanks very much for your comment. The damage by CTX in testis and epididymis of asthenospermia mouse model is not such severe, so the effect of CNP on tissue damage is not such clear.  But we can find difference in histology. Compared to CTX group, the number of spermatogonia were restored and the arrangement of germ cell became more regularly in testis and epididymis in CNP treatment groups (Fig 2B).

CNP belongs to the natriuretic peptide family which mediates its biological effects as a paracrine or autocrine regulator. El-Gehani F, et al. (Biol Reprod. 2001) reported that natriuretic peptides (ANP\BNP\CNP) all stimulated testosterone production, with significant effect at concentrations > or =1 x 10(-8) mol/L of ANP, > or =1 x 10(-9) mol/L of BNP, and > or =1 x 10(-6) mol/L of CNP. Knockdown of NPR2 by RNAi resulted in S phase cell cycle arrest, cell apoptosis, and decreased testosterone secretion in mouse Leydig cells (Yang L, et al. World J Mens Health. 2019.). Our results in figure 2E showed that CNP could increase the production of testosterone, which is consistent with the results of other researchs. This part has been added in the text and marked in blue.

  1. Mat and Met: line 274 to 284: totally un-understandable; not precise: what dye for ROS??

Reply: Thanks very much for your suggestion. This part has been revised and marked in blue. We use dihydroethidium (DHE) to dye for ROS. DHE can freely penetrate living cells, and is oxidized by intracellular ROS to form ethidium oxide; Ethidium oxide can be incorporated into chromosomal DNA to produce red fluorescence. Based on the production of red fluorescence in living cells, the amount and change of cellular ROS content can be judged.

Reviewer 2 Report

In this submission, the effects of CNP on the sperm motility was analysed with two experimental models, one on sperm from asthenospermic human and the second with an experimental infertile model mice.

In these two approaches, CNP is found to be active on the sperm motility and reduce damage in testis and epididymis induced by CTX.

General comments:

The aim of this study was to prove that CNP is active on deficient sperm motility. The experimental scheme of this study is found to be artificial. It should be better to analyse the effect of CNP on normal sperm and compare the results on sperm from asthenospermia. The experimental rat infertile seem to be another subject not only related to sperm deficient. The authors should dissociate their results in two submissions.

Author Response

Dear reviewer:

Thank you for your comments concerning our manuscript entitled “C- type natriuretic peptide (CNP) could improve sperm motility and reproductive function of asthenozoospermia” (ID: ijms-1860087). We have read these comments very carefully and found these suggestions are most helpful. Additionally, the manuscript was revised carefully according to the comments and marked in blue color. To be specific, we made revision as followed:

The aim of this study was to prove that CNP is active on deficient sperm motility. The experimental scheme of this study is found to be artificial. It should be better to analyse the effect of CNP on normal sperm and compare the results on sperm from asthenospermia. The experimental rat infertile seem to be another subject not only related to sperm deficient. The authors should dissociate their results in two submissions.

Reply: Thanks very much for your comment. Our previous study showed the CNP could improve sperm motility and fertility in normal people (published in Asian J Androl. 2016; Reprod Biomed Online. 2019). Therefore, we presumed the CNP could also improve the sperm motility in asthenozoospermia. The results of this study have confirmed our hypothesis. Because CNP couldn’t be added to the human body in vivo, we used the asthenospermia mouse model to explore the mechanism of CNP on sperm motility and reproductive function. Though the model couldn’t imitate completely the status in asthenozoospermia patient, it is still an effective way to study the mechanism. In this manuscript, we use in vitro and in vivo study to prove the effect of CNP on asthenospermia and explore the mechanism.

Round 2

Reviewer 1 Report

The main problems detected have been corrected